# Factors associated with treatment initiation delay among new adult pulmonary tuberculosis patients in Tigray, Northern Ethiopia

**Kiros Tedla** [1]*, **Girmay Medhin**[2], **Gebretsadik Berhe**[3], **Afework Mulugeta**[3], **Nega Berhe**[2]

**1** Institute of Biomedical Sciences, College of Health Sciences, Mekelle University, Mekelle, Ethiopia,
**2** Aklillu Lema Institute of Pathobiology, Addis Ababa University, Addis Ababa, Ethiopia, **3** School of Public Health, College of Health Sciences, Mekelle University, Mekelle, Ethiopia

* kirosmerry12@gmail.com

## Abstract

### Background

Delayed treatment initiation of Tuberculosis patients results in increased infectivity, poor treatment outcome, and increased mortality. However, there is a paucity of evidence on the delay in new adult pulmonary Tuberculosis patients to initiate treatment in Tigray, Northern Ethiopia.

### Objective

To assess the factors associated with treatment initiation delay among new adult pulmonary tuberculosis patients in Tigray, Northern Ethiopia.

### Methods

The study design was cross-sectional. A total of 875 new adult pulmonary tuberculosis patients were recruited from 21 health facilities from October 2018 to October 2019. Health facilities were selected by simple random sampling technique and tuberculosis cases from the health facilities were consecutively enrolled. Data were collected using structured questionnaire within the first 2 weeks of treatment initiation. Delay was categorized as patient, health system and total delays. Data were analyzed using SPSS version 21 and logistic regression was used to identify factors associated with the odds of delays to initiate treatment. A p-value of less than 0.05 was reported as statistically significant.

### Results

The median patient, health system and total delays were 30, 18 and 62 days, respectively. Rural residence, being poor, visiting non-formal medication sources, being primary health care and the private clinic had higher odds of patient delay whereas being HIV positive had lower odds of patient delay. Illiteracy, first visit to primary health care and private clinic had

**Data Availability Statement:** All relevant data are within the manuscript and its Supporting Information files.

**Funding:** The authors received no fund for this research work.

**Competing interests:** The authors declared that there is no any competing interest.

higher odds of health system delay whereas a visit to health facility one time and have no patient delay had lower odds of health system delay.

## Conclusion

The median patient delay was higher than the median health system delay before initiating treatment. Hence, improved awareness of the community and involving the informal medication sources in the tuberculosis pathways would reduce patient delay. Similarly, improved cough screening and diagnostic efficiency of the lower health facilities would shorten health system delay.

## Introduction

Tuberculosis (TB), caused by *Mycobacterium tuberculosis*, is one of the leading causes of morbidity and mortality worldwide and remains a major public health problem in many developing countries [1, 2].

Early diagnosis and prompt effective therapy are key elements of successful TB control. But, delayed presentation and treatment initiations are common in both low and high-income countries which may result in prolonged infectivity in the community [3, 4, 5]. A systematic review reported that the median time of delay from onset of cough until treatment initiation varied from 21–136 days [6]. This could be mainly due to the delay in the pathways of TB treatment such as the time elapsed in seeking health/patient delay or the time elapsed at the health facility due to the failure of the health system to diagnose the disease [7, 8]. It is estimated that untreated TB patient can infect an average of 10 contacts annually and over 20 during the natural course of the disease until death [7]. Delay in tuberculosis diagnosis may also lead to a more advanced disease stage at presentation, which contributes to late sequelae, poor treatment outcome and increased mortality [9, 10].

In Ethiopia, delay varies from 60–120 days from onset of symptoms till initiation of treatment [8, 11] with median patient delay of 20–90 days [8, 12, 13] and provider delay of 6–34 days [8, 13, 14]. Risk factors for delay include rural residence [15], lower educational level [16, 14], being women, large family size, and stigma [16], being old age [15], first visit to non-formal health providers [14], first visit to health posts [6] and form of TB [15, 16, 17].

Almost all the studies in Ethiopia focused on either health centers or general hospitals [13, 15, 18] and lacks coverage across all of the levels of the health system but this research was done across all the levels of health system including health centers, general hospitals and referral hospitals. Most of the previous studies were also conducted when the directly observed treatment short course (DOTS) regimen was given for eight months which was later changed into six months starting from 2011 [19]. Therefore, the present study was conducted to investigate delay to initiate treatment (patient and health system) and factors associated with delay among new PTB cases at health centers and hospitals in Tigray, Northern Ethiopia. This would enable for the planning of specific interventions focusing patient, health system or both types of delays that yields the maximum benefit for TB patients and the community at large.

## Methods and materials

### Study setting

This study was done in 16 health centers and 5 hospitals (4 general hospitals and one specialized referral hospital) from two zones of Tigray, Northern Ethiopia. Tigray Regional State is

located at 12o 15–14 o 57and 36˚27- 39˚59 latitude. Excluding Mekelle town, the state capital, there are six administrative zones in Tigray. The region has a total population of above 5.3 million which is about 6% of the total population of Ethiopia, over 80% live in the rural areas [20]. There are 214 health centers and 38 hospitals (primary hospital, general hospital, and specialized hospitals) across the region [21].

## Study design, participants and study period

A cross-sectional survey was conducted to investigate patient and health system delay of pulmonary TB (PTB) to initiate treatment in health facilities of two zones of Tigray region. Study participants were newly diagnosed bacteriologically positive or x-ray positive pulmonary tuberculosis cases, above 18 years of age, diagnosed in the study settings from October 2018 to October 2019. Retreated and multiple drug resistance (MDR) TB patients were excluded from the study.

## Sample size

The sample size required to estimate the magnitude of delay for initiation of treatment was determined taking an estimate of 81.5% reported in the previous study for the proportion of delay of more than one month [13], 95% confidence in the estimate and 2.6% margin of error. This resulted in a sample size of 850. Hence, including 3% non-response rate, the total sample size was 875.

## Sampling procedure

Two zones (i.e. Eastern zone and Mekelle zone) were selected for the current study among the seven zones of the region. These two zones are among the highly populated zones of the region. These zones were selected because of the nature of the research project which needs frequent supervision, follow up and are close to the research Institute where the TB culture is performed. Furthermore, the only functional specialized hospital is found only in Mekelle and the two zones can also represent the region. Within the selected zones, there were 26 primary health care units and 6 hospitals that had adopted DOTS and reported average monthly cases of 3 and above according to the 2016/2017 and 2017/18 HMIS report of Regional Health Bureau [21]. Of these, 21 health facilities (16 health centers and 5 hospitals including the specialized hospital) were included in this study. Health centers were selected by a simple random sampling (SRS) method (lottery method) from the list of the health centers but among six hospitals one hospital was excluded as it is working as ophthalmological center only. A target number of interviewees per facility was proportionally allocated based on the average number of TB cases reported by the facility in prior two years and were consecutively included in the study until the optimal sample size was obtained (see the sampling procedure in Fig 1).

## Data collection instruments and procedures

Data were collected using questionnaire developed for this study by adopting from questionnaires used in previous studies [7, 8, 11, 22]. Trained data collectors interviewed study participants in the first 2 weeks of their treatment, after obtaining their informed consent. The questionnaire included information about socio-demographic and socio-economic characteristics, lifestyle factors like smoking, alcohol consumption, health-seeking behavior, the time since the first symptom, action taken to symptom and others. The study participants were also asked questions regarding the elements that might influence their health-seeking behavior, such as fear of what would be found on diagnosis, fear of social isolation, stigma, and

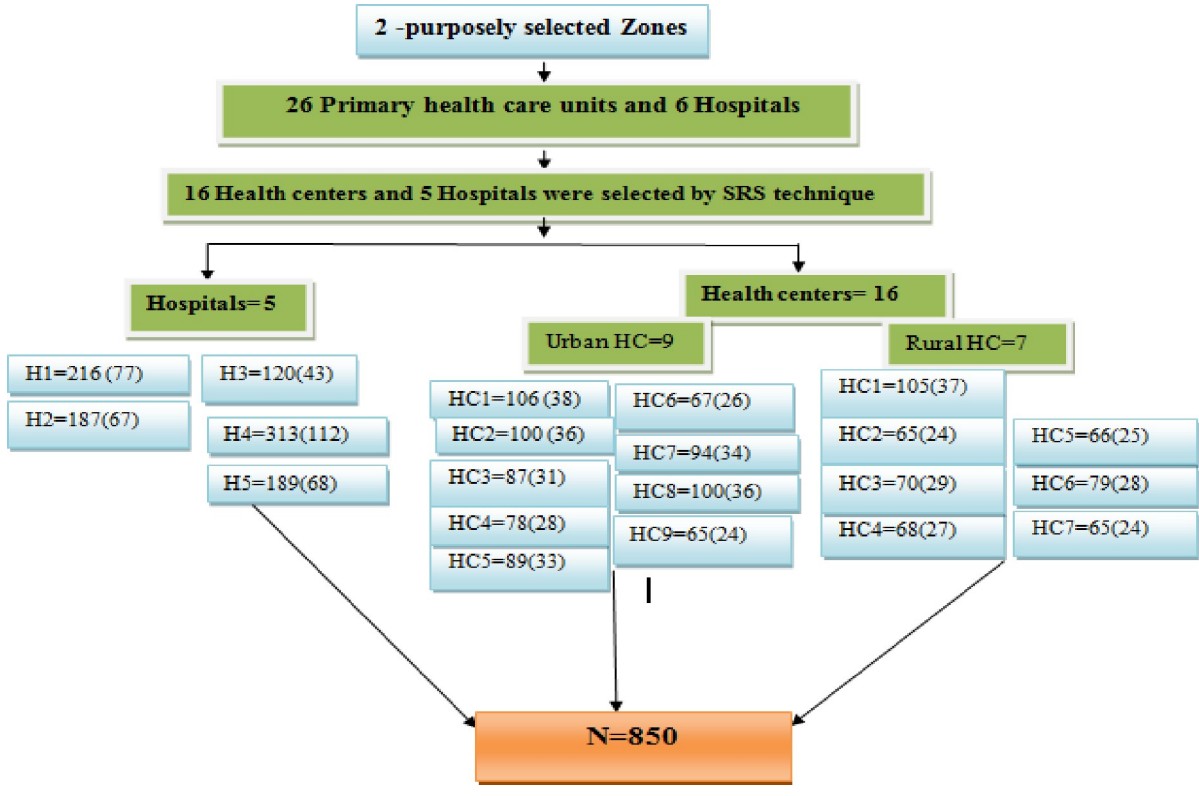

**Fig 1. Showing sampling procedure of the study settings and study participants.**

knowledge regarding the disease, satisfaction with care and other issues like distance from the health facility.

## Data quality assurance

The questionnaire was pre-tested on 50 TB patients recruited from all of the study settings. During this pre-testing content reliability were assessed. Test-retest reliability was assessed by giving the questionnaire to the same person twice at an interval of 2 weeks and measuring the intra-class correlation coefficient to evaluate the intra-rater reliability. Re-interviewing the same person by two different interviewers and measuring the intra-class correlation coefficient was also performed to measure the inter-rater reliability. The adapted questionnaire was reviewed by experts in Tigray Regional Health Bureau and other researchers experienced in this field to clarify confusing items and to comment on the apparent validity of each item. The comments from the expertise were included and the resulting questionnaire was administered to data collectors to further check items that might not be easy to understand. Finally, the questionnaire was modified according to the results of the pilot phase and the comments from the experts. The final questionnaire was used for data collection. A total of 21 nurses with BSc degrees were used as the data collectors of which three were senior supervisors. These data collectors were trained on the objectives of the study, the importance of the findings and on how to collect the data from study participants. Close supervision was also conducted during the data collection time. The questionnaire was converted into local language (Tigrigna) and the entire interview was conducted using the local language.

## Data analysis

Data were entered and processed using SPSS version 21. Proportion for categorical variables and, mean/median for continues variables were used to summarize the collected data. The normality of numeric variables was assessed using plots (Q-Q plots, P-P plots, and histograms) and Kolmogorov-Smirnov test. Following this process, the distribution of the number of days elapsed across different pathways of the delay was not normally distributed. Thus median patient delay and provider delay were compared across different binary groups using Mann-Whitney U and using Kruskal Wali's tests across variables with more than two categories. To dichotomize the sample into either shorter or longer delay period we took 30 days as a cut off for patients' and 15 days as a cut off for health systems' delay. Thirty days was chosen as it was the median for patients' delay but it was also due to the reason that TB patients who had remained untreated for one month since showing clinical signs would have aggravated clinical outcomes [9, 23]. However, the decision to use 15 days was based on the consultation of treating physicians to TB suspected patients and using the experience of previous Ethiopian studies [8]. Finally, logistic regression models were fitted to identify factors associated with the probability of delay to patient and health system. The selection of the variables for multiple regression was made based on p-value ≤0.2 on crude analysis. The goodness of fit of the logistic regression model was checked using the Hosmer and Lemeshow test. In all the statistical tests, any result with the p-value less than 5% was considered as statistically significant.

## Ethical consideration

Ethical approval was obtained from the Research and Ethical Committee of Addis Ababa University, Aklillu Lemma Institute of Pathobiology. A formal permission letter was also obtained from the Regional Health Bureau of Tigray and submitted to the selected district health offices. Each health facility selected for the study was contacted with a permission letter from the district health office. Each case diagnosed as TB according to the national guideline consented in a written form before the interview. To assure confidentiality, interview with TB case was held in a private room and the information collected was recorded anonymously.

## Operational definitions

**Total delay**: the time from the onset of TB symptoms to the first start of anti-TB treatment.

 **Long total delay**: if the time from the onset of TB symptoms to the first start of anti-TB treatment is more than 30 days.

 **Patient delay**: the time from the onset of symptoms until the first contact with the health care service provider in both private and governmental health facilities.

 **Health system delay**: the time from a patient's first contact with the health care service provider until either a TB diagnosis is made or TB treatment is commenced in both private and governmental health facilities.

 **Diagnostic delay**: the time from first contact with health care service provider until a diagnosis is made.

 **Treatment delay**: the time from when a diagnosis of TB is made until treatment is started.

## Result

### Socio-demographic characteristics of the study participants

A total of 875 study subjects were participated in this study. Of the study participants, 58.1% were males, and 54.5% urban residents. Furthermore, 27.2% were farmers, 21.6% housewives,

18.9% employed, 10.4% daily laborers, 11.4% students, and 10.5% unemployed. The mean age was 38.2 years (sd = 15.1) (Table 1).

## Clinical characteristics of the study participants

The mean (sd) BMI of the study subjects during treatment initiation was 17.9±2.7 kg/m$^2$. The major clinical symptoms that were experienced by the patients were cough (99.2%), fever (48.6%), and weight loss (45.5%). Regarding HIV status, 11.3% were HIV positive while 2.6% of the study participants had other chronic diseases. About 36.6%, 88.2% and 45.5% of the study participants were AFB, GeneXpert and X ray positive, respectively (Table 2).

## Health accessibility and health seeking behavior

More than half of the study participants (57.0%) live within 10km distance to nearest health facility. Furthermore, 64.8% of the study participants visited more than one health facility

**Table 1. Socio-demographic and lifestyle characteristics of new adult pulmonary tuberculosis patients attending selected health facilities of Tigray, Northern Ethiopia, 2019(n = 875).**

| Variable | Categories | Frequency | Percent |
|---|---|---|---|
| Age, years | 18–25 | 228 | 26.1 |
| | 26–44 | 392 | 44.8 |
| | 45–54 | 144 | 16.5 |
| | ≥55 | 111 | 12.7 |
| Sex | Male | 508 | 58.1 |
| | Female | 367 | 41.9 |
| Religion | Orthodox | 804 | 91.9 |
| | Muslim | 54 | 6.2 |
| | Others (catholic and protestant) | 17 | 1.9 |
| Educational status | No formal education | 349 | 39.9 |
| | Primary school | 325 | 37.1 |
| | Secondary school | 109 | 12.5 |
| | College and above | 92 | 10.5 |
| Occupation | Farmer | 238 | 27.2 |
| | Housewives | 189 | 21.6 |
| | Employed | 165 | 18.9 |
| | Daily laborer | 91 | 10.4 |
| | Student | 100 | 11.4 |
| | Unemployed | 92 | 10.5 |
| Income | Poor (Indebt) | 328 | 37.5 |
| | Medium (Income = expense) | 321 | 36.7 |
| | Saving (rich) | 226 | 25.8 |
| Marital status | Married | 570 | 55.1 |
| | Divorced/Widowed | 71 | 8 |
| | Single | 234 | 26.7 |
| Residence | Rural | 398 | 45.5 |
| | Urban | 476 | 54.5 |
| History of smoking | Yes | 89 | 10.2 |
| | No | 786 | 89.8 |
| Alcohol use | Yes | 464 | 53 |
| | No | 411 | 47 |

**Table 2. Clinical characteristics of new adult pulmonary tuberculosis patients attending selected health facilities of Tigray, Northern Ethiopia, 2019 (n = 875).**

| Variable | | Frequency | Percent |
|---|---|---|---|
| **Clinical signs and symptoms** | Cough | 868 | 99.2 |
| | Fever | 425 | 48.6 |
| | Chest pain | 389 | 44.5 |
| | Weight loss | 398 | 45.5 |
| | Haemoptysis | 207 | 23.7 |
| | Others[a] | 449 | 51.3 |
| **Presence of other chronic diseases[b] (excluding HIV)** | Yes | 23 | 2.6 |
| | No | 852 | 97.4 |
| **HIV status** | Positive | 99 | 11.3 |
| | Negative | 776 | 88.7 |
| **Diagnosis** | **AFB** | | |
| | Positive | 320 | 36.6 |
| | Negative | 555 | 63.4 |
| | **GeneXpert** | | |
| | Positive | 772 | 88.2 |
| | Negative | 103 | 11.8 |
| | **X-ray** | | |
| | Positive | 398 | 45.5 |
| | Negative | 477 | 54.5 |

[a] Sweating, loss of appetite, fatigue, chill, malaise

[b] diabetes, arthritis, epilepsy, chronic liver disease, Visceral leishmaniasis, COPD

before the actual diagnosis. Regarding their first action to the current illness, 78.6% visited health care provider (HCP), and the major symptoms which made the study participants to visit HCP were cough and haemoptysis. About 43.8% had good knowledge about TB, 45.6% had low degree to stigma and 49% had adequate satisfaction with care (Table 3).

## Delay of TB patients to initiate treatment

Patients took a mean of 86 days and a median of 30 (IQR: 11–180) days to first visit a health care service provider after the onset of their symptoms. Similarly, the mean and median number of days patients delay from the first visit to the health care service provided to the TB diagnosis or initiation of TB treatment in the health system was 39 and 18 days with inter quartile range (IQR) of 2–72 days, respectively. The median (IQR) delay in diagnosis and treatment were 15(1–40) and 2(1–3) respectively. The mean and median numbers of days patients spent to come to health facilities and get the treatment (total delay) were 124 and 62 days, with interquartile range of 16–221 days respectively (Fig 2).

Of the entire study participants, 55.2% visited health facility within 30 days of their first sign and symptom. Of all the study participants 52.7% had got the diagnosis and treatment in the health facility within 14 days of their first visit whereas 47.3% took more than 14 days to get the diagnosis and treatment in the health facility. Of all the study participants 26.2% came to health facility and got diagnosis and treatment within 30 days; of the total delayed patients, 52% were due to the patient delay.

**Factors associated with patient delay.** Summary results from bivariate and multivariable logistic regression showed that those who lived in rural area (OR = 18.7, 95%CI 12.2–28.7), being poor (being indebt) (OR = 5.5, 95%CI 3.3–9.2) and having medium economic status

**Table 3. Health accessibility and health seeking behavior of new adult pulmonary tuberculosis patients attending selected health facilities of Tigray, Northern Ethiopia, 2019 (n = 875).**

| Variable | | Frequency | Percent |
|---|---|---|---|
| **Distance from the nearest health facility** | ≤10 | | |
| | >10 | | |
| | | 499 | 57 |
| | | 376 | 43 |
| **Health facility visits to the current illness** | 1 | 308 | 35.2 |
| | 2 and above | 567 | 64.8 |
| **First action to the current illness** | HCP | 688 | 78.6 |
| | Holly water | 79 | 9.0 |
| | Traditional healer | 43 | 4.9 |
| | Self medication | 65 | 7.4 |
| **Type of health facility first visited** | Hospital | 367 | 42 |
| | Health center | 332 | 38 |
| | Private facility | 175 | 20 |
| **Type of symptom made to seek HCP** | Cough | 531 | 60.7 |
| | Fever | 53 | 6.1 |
| | Chest pain | 143 | 16.3 |
| | Haemoptysis | 148 | 16.9 |
| **Knowledge about TB** | Good (> median) | 383 | 43.8 |
| | Poor (< median) | 492 | 56.2 |
| **Stigma** | High (> median) | 399 | 45.6 |
| | Low (< median) | 476 | 54.4 |
| **Satisfaction** | Good (> median) | 429 | 49 |
| | Low (< median) | 446 | 51 |
| **Reasons for consultation HCP** | Confidence in getting cured | 275 | 35.2 |
| | Accessible | 210 | 26.9 |
| | Referred by health facility | 117 | 15 |
| | Others[c] | 162 | 20.8 |
| **Reasons for non-consultation to HCP** | Too far | 50 | 48.1 |
| | Bad experience | 26 | 25 |
| | Others[d] | 24 | 23.1 |
| **Perceived cause of patient delay** | Poor staff attitude | 361 | 48.3 |
| | Poor quality of health facility | 139 | 18.6 |
| | Hoped their symptom would go away by its own | 72 | 9.6 |
| | Others[e] | 114 | 15 |

[c] advised by some body and service available any time

[d] too busy, insufficient medication, long waiting time and the patient did not consider it important

[e] fear of social, economic constraints, fear of what would be found on diagnosis, poor quality of care and work load

(OR = 2.6, 95%CI 1.6–4.4), visiting to non-formal medication sources (OR = 7, 95%CI 4.3–11.5), being primary health care (OR = 1.6, 95%CI 1.1–2.4) and private clinic (OR = 1.7, 95%CI 1–2.8) were significantly associated with increased patients' delay. Whereas HIV positive individuals were less likely to delay compared to those HIV negative individuals (OR = 0.52, 95%CI 0.26–0.99) (Table 4).

**Factors associated with health system delay.** Findings from the bivaraite and multivariable logistic regression analysis showed that those who were illiterate (OR = 2.12, 95%CI 1.27–3.56), those with secondary school education (OR = 1.73, 95%CI 1.03–2.89), first sought in

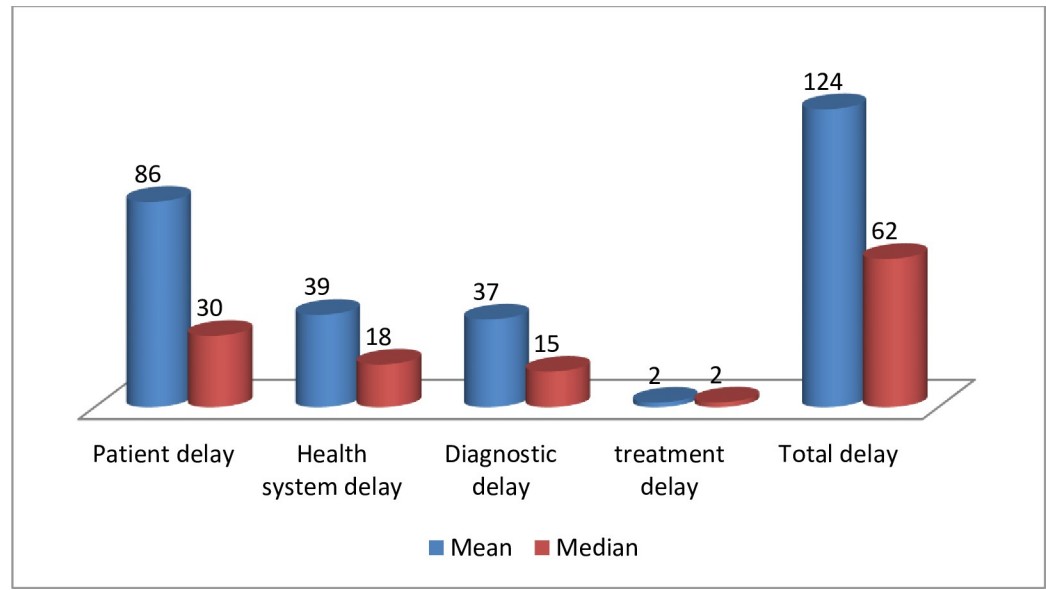

**Fig 2. Mean and median delays of the study participants during their health seeking pathways in Tigray, Northern Ethiopia, October 2018 to October 2019.**

primary health care (OR = 1.7, 95%CI 1.3–2.3) and private clinic (OR = 1.7, 95%CI 1.2–2.5) were significantly associated with increased health system delay. Whereas those who visited

**Table 4. Factors associated with patient delay among new adult pulmonary tuberculosis patients attending selected health facilities of Tigray, Northern Ethiopia, 2019 (n = 875).**

| Variables | Categories | Frequency | | COR (95%CI) | AOR (95%CI) |
|---|---|---|---|---|---|
| | | Patient delay ≤30 days | Patient delay >30 days | | |
| **Sex** | Male | 193 | 174 | 1.00 | 1.00 |
| | Female | 290 | 218 | 1.2(0.92–1.6) | 1.1(0.64–1.8) |
| **Income** | Poor (Indebt) | 118 | 210 | 8.8(5.8–14) | 5.5(3.3–9.2)* |
| | Medium (income = expense) | 177 | 144 | 4(2.7–6.1) | 2.6(1.6–4.4)* |
| | Rich (Saving) | 188 | 38 | 1.00 | 1.00 |
| **Residence** | Rural | 86 | 306 | 15(10.8–21) | 11(7.6–16)* |
| | Urban | 390 | 92 | 1.00 | 1.00 |
| **Type of health facility** | Hospital | 233 | 134 | 1.00 | 1.00 |
| | Health center | 164 | 168 | 1.8(1.3–2.4) | 1.6(1.1–2.4)* |
| | Private clinic | 86 | 89 | 3(1.3–2.6) | 1.7(1–2.8)* |
| **Distance from nearest health facility, km** | ≤10 | 287 | 210 | 0.79(0.6–1) | 0.8(0.54–1.2) |
| | >10 | 195 | 181 | 1.00 | 1.00 |
| **Satisfaction** | Good | 308 | 138 | 0.3(.23–.41) | 0.9(0.56–1.3) |
| | Poor | 175 | 254 | 1.00 | 1.00 |
| **First action to symptoms** | Health facility visit | 447 | 241 | 1.00 | 1.00 |
| | Visit to non formal | | | | |
| | medication source | 36 | 151 | 7.8(5.2–11.5) | 7(4.3–12)* |
| **HIV status** | Positive | 64 | 35 | 0.6(0.42–0.99) | 0.52(0.26–0.99)* |
| | Negative | 419 | 357 | 1.00 | 1.00 |

*statistically significant at p<0.05

**Table 5. Factors associated with health system delay among new adult pulmonary tuberculosis patients attending selected health facilities of Tigray, Northern Ethiopia, 2019 (n = 875).**

| Variables | Categories | Frequency | | COR | AOR |
|---|---|---|---|---|---|
| | | Health system delay ≤15 | Health system delay >15 | | |
| **Type of diagnosis AFB** | Yes | 138 | 182 | 1.8(1.4–2.4) | 0.75(0.47–1.2) |
| | No | 323 | 232 | 1.00 | 1.00 |
| **GeneXpert** | Yes | 397 | 375 | 1.6(1–2.4) | 1.2(.76–2) |
| | No | 64 | 39 | 1.00 | 1.00 |
| **X-ray** | Yes | 176 | 222 | 1.9(1.4–2.5) | 1.3(0.87–2) |
| | No | 285 | 192 | 1.00 | 1.00 |
| **Educational status** | Illiterate | 184 | 165 | 2.1(0.91–2.9) | 2.1(1.3–3.6)* |
| | Primary school | 156 | 169 | 1.6(0.91–2.9) | 1.4(.76–2.6) |
| | Secondary school | 60 | 49 | 1.8(1.1–2.85) | 1.7(1–2.9) |
| | College and above | 61 | 31 | 1.00 | 1.00 |
| **Type of health facility first sought** | Public hospital | 225 | 142 | 1.00 | 1.00 |
| | Health center | 160 | 172 | 1.7(1.3–2.3) | 1.3(1–1.9)* |
| | Private clinic | 84 | 91 | 1.7(1.2–2.5) | 1.5(1–2.2)* |
| **HIV status** | Positive | 54 | 45 | 0.91(0.60–1.4) | 0.91(.56–1.5) |
| | Negative | 407 | 369 | 1.00 | 1.00 |
| **Number of health facility encounters** | 1 | 204 | 104 | 0.26(0.18– | 0.21(0.2–0.46)* |
| | >1 | 257 | 310 | 0.37) 1.00 | 1.00 |
| **Patient delay** | Yes | 157 | 235 | 1.00 | 1.00 |
| | No | 304 | 179 | 0.39(0.3–0.52) | 0.54(0.39–0.76)* |

*statistically significant at p<0.05

health facility one time (OR = 0.21, 95%CI 0.20–0.46) and those who didn't come delayed at their first presentation after developing the signs and symptoms (OR = 0.54, 95%CI 0.39–0.76) were less likely to delay compared to their counter parts (Table 5).

## Discussion

The median patient, health system and total delays were 30 days, 18 days and 62 days in the study setting respectively. Whereas the proportion of TB patients patient, health system and total delays were 55.2%, 52% and 73.8% respectively. The determinant factors affecting patient delay were HIV status, being health center or private clinic, visiting non-formal medication sources, and economic status. Absence of formal education, repeated visit to lower health facilities, delayed presentation to health facility and first sought at the health center and private clinic were associated with health system delay.

### Length of delay in the pathways to initiate TB treatment

The median patient and health system delays in the current study were similar with previous findings reported in Ethiopia [9, 17] and Africa [24, 25]. However, the median patient delay was shorter than previous studies in Ethiopia [16, 26, 27] and longer than other reports from Ethiopia [28, 29] and African [3, 30] but the health system delay was shorter than studies in Ethiopia [13, 14] which have similar median patient delay with this study. The reason for the shorter delay might be related to while two of the previous studies [16, 29] were done in pastoralist communities; the current study was done in the agrarian community having relatively better accessibility and knowledge about TB compared to the pastoralist community. The

reason for the longer patient delay compared to other African countries could be related with accessibility to the health facility as about 87.5% and 89% of the study participants in Tanzania and Chad had lived with less than 10km from the health facility respectively, while in this study only 57% live with a distance of less than 10km from the health facility [3, 30].

## Factors linked with patient-related delays to initiate TB treatment

Rural residents had higher odds of patient delay compared to the urban residents in health-seeking which was supported by studies done in Ethiopia [27, 31, 32], and Mediterranean region [7]. This might be related to the poor access of rural residents to health information and health facilities compared to the urban residents. Study had indicated that rural residency makes it difficult to travel to health facilities in terms of travel time from patients' areas of residences to public facilities [33].

Being low in economic status was associated with increased patient delay which is supported by studies from Ethiopia [26], Africa [34] and Asia [35]. This might be related to the patients' inability to cover the direct and indirect costs like transport, food, consultations, investigation costs like X-ray and others even though laboratory (GeneXpert and AFB) and treatment costs are free. A study from Kenya had indicated that patients spend 7.1% of their median household income before actual diagnosis and in Ethiopia patients spend 125% of their monthly income to get a proper diagnosis [36, 37] which might explain the delayed presentation of the poor to the health facility.

Visiting non-formal medication sources (holy water, traditional medication, and self-medication) were strongly associated with delayed presentation of patients to the health system. This is in line with studies from Ethiopia [13, 18, 7, 10, 14] and Africa [7]. This could be due to the use of some home remedies or antibiotics might lessen or mask the manifestation of the illness which might inhibit the timely presentation of the patient to the health care system.

Being primary health center and private clinic was associated with higher odds of patient delay. This is similar to studies in Ethiopia [17, 18], and Africa [33] but contrary to study from Ethiopia [30]. This is mainly due to low reliance or confidence of the patients on the lower health facilities as only about 35.2% of the health facility consulted patients had confidence in curing their illness at the lower health facilities (health centers) in this study.

However, being HIV positive was inversely related to patient delay as HIV positive patients were presented to the health facility earlier than HIV negative TB patients. This study was contrary to researches done in Ethiopia [19] but similar to study from Asia [38]. This could be by the fact that HIV-positive TB patients had more serious clinical symptoms which may have prompted them to seek treatment earlier [12, 38].

## Factors linked with health system-related delays to initiate TB treatment

Absence of formal education was strongly associated with health system delay which is similar to studies done in Ethiopia [19, 27], and Africa [7]. Those with college and above education levels might have better information about TB, increased awareness of TB disease which might help to seek care early and inform the health professional in a better way during diagnosis [39, 40].

Repeated visits to the lower health facilities (health centers and private clinic) were strongly associated with health system delay which is similar with studies done in Ethiopia [17], and Africa [24]. This might be due to the poor identification of TB patients by the health professionals as indicated in a recent study from Ethiopia where one third and nearly half of health care workers of public health facilities had poor knowledge and unsatisfactory practice on the management of tuberculosis infection [39].

Patients who were delayed during presentation were also more likely to have a delay in the health system; which is similar to studies from other parts of Ethiopia [17] but contrary to study from Georgia [40]. The reason for such difference might be related to the cut-off points used to ascertain delayed and non-delayed patients. Furthermore, the reason for such delay would be due to patients use different self medications and homemade remedies which might affect altering the clinical manifestations of the patients [40]. Furthermore, studies had also reported that more severe disease at presentation [23] hinders timely [14] diagnosis.

TB patients who first sought service in the health center and private clinic were associated with higher odds of health system delay. This is similar to other studies from Ethiopia [17, 18], and Africa [33] but contrary to study from Ethiopia [30]. This is mainly due to the lower health facility and private clinic had the luck of adherence to guideline, trained personnel and equipment specific for TB diagnosis. Furthermore, currently in the area where this study was conducted all hospitals are equipped with GeneXpert. The current research had limitations like measurement of delays relied on patient self-report which is liable to recall bias. However, we tried to minimize this bias through interviewing patients soon after diagnosis and helping them to recall using local events like national holidays, religious days, and dates of some events (birthdays, death, epiphany or marriage).

## Conclusion and recommendation

In general, more than thee forth of TB patients had initiated treatment after a delay of more than a month where both patient and health system delays had contributed equally in the study area. Moreover, the median delay elapsed during patients' presentations to health provider was higher compared to the time elapsed at the health system. Rural residents, being poor and medium income, visiting non-formal medication sources and being primary health care or private clinic were associated with increased patient delay whereas, being HIV positive was inversely related to the patient delay. Illiteracy, first visit to the primary health care and private clinic, repeated visit to the lower health facilities, and Patient delay were more likely to have a delay at the health system. Therefore, improving accessibility to diagnosis mainly to the rural residents through implementation of rapid diagnostic tools at the lower health facility, screening every individual with cough regardless of the time the cough starts and strengthening the public-private partnership in diagnosis and treatment of TB patients in order to reduce both patient and health system delay. Implementing strategies to collaborate with informal medical sources (traditional healers, local drug sellers, and religious leaders) on how to suspect TB symptoms & refer to government health facilities for diagnosis is recommended.

## Supporting information

**S1 Data.**
(XLSX)

**S1 File.**
(DOCX)

## Acknowledgments

We would like to express our gratitude to Aklilu Lemma Institute of Pathobiology, Addis Ababa University, for giving us the opportunity to undertake research. We would like also to thank Mekelle University for giving us full sponsorship to conduct this study. We would also like to thank all health professionals working in the health facilities where this study was conducted, the study participants and Tigray regional health bureau.

## Author Contributions

**Conceptualization:** Kiros Tedla, Girmay Medhin, Afework Mulugeta, Nega Berhe.

**Data curation:** Kiros Tedla, Girmay Medhin, Gebretsadik Berhe, Afework Mulugeta.

**Formal analysis:** Kiros Tedla, Girmay Medhin, Afework Mulugeta.

**Methodology:** Kiros Tedla, Girmay Medhin, Gebretsadik Berhe, Afework Mulugeta, Nega Berhe.

**Supervision:** Girmay Medhin, Gebretsadik Berhe.

**Visualization:** Kiros Tedla, Gebretsadik Berhe, Afework Mulugeta, Nega Berhe.

**Writing – original draft:** Kiros Tedla, Nega Berhe.

**Writing – review & editing:** Kiros Tedla, Girmay Medhin, Gebretsadik Berhe, Afework Mulugeta, Nega Berhe.

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
