## [Decision Letter · Decision Letter 0]

11 Feb 2020

PONE-D-19-35255

Patient and health system delay to initiate tuberculosis treatment and associated risk factors among pulmonary tuberculosis patients in Tigray, Northern Ethiopia: A multi-center institutional based cross sectional study

PLOS ONE

Dear Mr. Gebrehiwot,

Thank you for submitting your manuscript to PLOS ONE. After careful consideration, we feel that it has merit but does not fully meet PLOS ONE’s publication criteria as it currently stands. Therefore, we invite you to submit a revised version of the manuscript that addresses the points raised during the review process.

We would appreciate receiving your revised manuscript by Mar 27 2020 11:59PM. To enhance the reproducibility of your results, we recommend that if applicable you deposit your laboratory protocols in protocols.io, where a protocol can be assigned its own identifier (DOI) such that it can be cited independently in the future. For instructions see: http://journals.plos.org/plosone/s/submission-guidelines#loc-laboratory-protocols

We look forward to receiving your revised manuscript.

Kind regards,

Elizeus Rutebemberwa

Academic Editor

PLOS ONE

Journal Requirements:

1. Please include additional information regarding the survey or questionnaire used in the study and ensure that you have provided sufficient details that others could replicate the analyses. For instance, if you developed a questionnaire as part of this study and it is not under a copyright more restrictive than CC-BY, please include a copy, in both the original language and English, as Supporting Information.

4. Please include your tables as part of your main manuscript and remove the individual files. Please note that supplementary tables (should remain/ be uploaded) as separate "supporting information" files

Reviewers' comments:

Reviewer's Responses to Questions

**Comments to the Author**

1. Is the manuscript technically sound, and do the data support the conclusions?

Reviewer #1: Yes

Reviewer #2: Yes

2. Has the statistical analysis been performed appropriately and rigorously? 

Reviewer #1: Yes

Reviewer #2: Yes

3. Have the authors made all data underlying the findings in their manuscript fully available?

Reviewer #1: Yes

Reviewer #2: Yes

4. Is the manuscript presented in an intelligible fashion and written in standard English?

Reviewer #1: Yes

Reviewer #2: Yes

5. Review Comments to the Author

Reviewer #1: 1. Better to short the topic. Its too long.

2. data analysis and interpretation is in the satisfactory level.

3. conclusion is based on the results.

4. there is scientific value and it will add new data to the survey of TB

Reviewer #2: The article is interesting mainly because it describes as it's works the TB assistance in Northern Ethiopia. But it has some aspects needed to be corrected.

1: IT'S TOO LONG!.

A) Introduction: the last paragraph isn't correct. B) Methods and Materials: Very long. "Study setting" long, long...; C): Results: Tables numbers are bad located at the "profile of sttudy participants"; D: Discussion: too long; E: Figures: ok; E: Tables: Table 1: the variable "Family size" it's not well located as a "Mean +- SD", can't to stay were it is.

6. PLOS authors have the option to publish the peer review history of their article (what does this mean?). If published, this will include your full peer review and any attached files.

Reviewer #1: Yes: Chamila Priyangani Adikaram

Reviewer #2: Yes: M.N. Altet Gomez

---

## [Author Response · Author response to Decision Letter 0]

13 Mar 2020

I would like to appreciate the reviews for their constructive and thoughtful comment on the manuscript. I have lean a lot from their comments. I would also like to thank the journal and journal editor for considering my work and giving me all the directions on how we can make the manuscript interesting to readers. 

Regarding the manuscript we have tried to incorporate the comments and suggestions given from the reviewers and the academic editor. Thank you again for your time.

---

## [Decision Letter · Decision Letter 1]

16 Jun 2020

Factors associated with treatment initiation delay among new adult pulmonary tuberculosis patients in Tigray, Northern Ethiopia

PONE-D-19-35255R1

Dear Dr. Gebrehiwot,

We’re pleased to inform you that your manuscript has been judged scientifically suitable for publication and will be formally accepted for publication once it meets all outstanding technical requirements.

Kind regards,

Frederick Quinn

Academic Editor

PLOS ONE

Additional Editor Comments (optional):

Reviewers' comments:

Reviewer's Responses to Questions

**Comments to the Author**

1. If the authors have adequately addressed your comments raised in a previous round of review and you feel that this manuscript is now acceptable for publication, you may indicate that here to bypass the “Comments to the Author” section, enter your conflict of interest statement in the “Confidential to Editor” section, and submit your "Accept" recommendation.

Reviewer #2: All comments have been addressed

Reviewer #3: (No Response)

2. Is the manuscript technically sound, and do the data support the conclusions?

Reviewer #2: Yes

Reviewer #3: Yes

3. Has the statistical analysis been performed appropriately and rigorously? 

Reviewer #2: Yes

Reviewer #3: Yes

4. Have the authors made all data underlying the findings in their manuscript fully available?

Reviewer #2: Yes

Reviewer #3: Yes

5. Is the manuscript presented in an intelligible fashion and written in standard English?

Reviewer #2: Yes

Reviewer #3: Yes

6. Review Comments to the Author

Reviewer #2: Your corrections are appropiate. Mainly you have shortened it. I think it may be usefool to have a best control ob TB.

Reviewer #3: Authors adequately address comments by previous reviewers

Comments:

Operational definitions: Health system delay in your manuscript is refers to both private and government health providers. Please, include in your operation definition that patient health facility visit is also include private health providers.

Study setting: Line 65: 12o 15 - 4 o 57 longitude. Please, check it for error. I think it is 12o 15 - 14 o 57 longitude.

Acknowledgements: Line: 325-329 please, revise this section.

7. PLOS authors have the option to publish the peer review history of their article (what does this mean?). If published, this will include your full peer review and any attached files.

Reviewer #2: No

Reviewer #3: No

---

## [Editor Report · Acceptance letter]

5 Aug 2020

PONE-D-19-35255R1 

Factors associated with treatment initiation delay among new adult pulmonary tuberculosis patients in Tigray, Northern Ethiopia 

Dear Dr. Gebrehiwot:

I'm pleased to inform you that your manuscript has been deemed suitable for publication in PLOS ONE. Congratulations! Your manuscript is now with our production department. 

Kind regards, 

on behalf of

Dr. Frederick Quinn 

Academic Editor

PLOS ONE